# The Impact of Managers’ Environmental Cognition on Urban Public Service Innovation from the Perspective of Green Ecology

**DOI:** 10.3390/ijerph192315945

**Published:** 2022-11-29

**Authors:** Ling Liu, Yuanyuan Zhou

**Affiliations:** 1School of Tourism, Xinyang Normal University, Xinyang 464000, China; 2School of Business Administration, Zhongnan University of Economics and Law, Wuhan 430073, China; 3School of Business, Xinyang Normal University, Xinyang 464000, China

**Keywords:** green ecology, environmental cognition, urban public service system, system building

## Abstract

With the continuous improvement in urban managers’ cognition of green ecology, it is a hot issue in current research to explore the role of managers’ cognition on urban public service innovation from the perspective of green ecology. The main purpose of this study was to explore ways to build green ecological cities and provide high-quality urban public services based on managers’ environmental cognition. Through sorting out and discriminating the concepts related to green ecology, this research improves the current theoretical system related to green ecological city services. A theoretical model of a green ecological city public service system was constructed, and its influence path and effect on green ecological city public service innovation were analyzed in detail. This research provides a good tool and method for follow-up research to better understand the composition and innovation of green ecological city public service systems.

## 1. Introduction

A city is the carrier of social development nowadays. With the acceleration of global urbanization, there are increasing requirements for urban public services. Green ecological urban construction and public services have become a current development trend. In this context, whether city managers have formed effective environmental cognition and whether they can provide green ecological public services for the city have not yet been concluded. Starting from the construction of a green ecological city public service system, this paper explores the effect path of city managers’ cognition on it through improving the system.

For city managers, the formation of their environmental cognition is accompanied by the deepening of the concept of green ecology. Regarding how to manage a city and provide public services, one of the earliest studies conducted in the 19th century gathered a large amount of information on urban population and industry convergence for the living environment of urban residents. This produced more serious challenges in order to explore effective, appropriate, and sustainable urban development. In many cities, the management development concept arose at a historic moment [1]. For example, the concept of a “garden city” in the early period and the concepts of an “ecological city”, a “healthy city”, and a “green city” have been put forward by subsequent scholars around the world. In recent years, with the rise of the green concept, the global urbanization wave has once again pushed the concepts of a “low-carbon city”, a “low-carbon eco-city”, and a “green eco-city” to a new research frontier [2]. Regarding urban management and public service provision under the concept of a “green ecological city”, most scholars believe that it is a sociological construct that includes both natural ecology and human ecology, requiring managers to pay attention to the protection of natural resources and the environment in the process of urban management [3]. At the same time, attention should be paid to adding more practical content such as “green life” and “green culture” in the process of providing public services for urban residents [4].

The main contribution of this paper lies in the in-depth analysis of managers’ environmental cognition on the management planning of an urban spatial layout, including the infrastructure facilities, housing, transportation, ecology, green space, and other public services. At the same time, by building a green ecological city public service system model, the current green ecological context connotation can be enriched and the extension of urban public service supplies—namely, in the city of the carrier—through a public service supply can achieve a political, economic, cultural, social, and ecological civilization construction of “Five-one project”, promote harmony between humans and nature as well as society, the economy, and building a community of life to realize the green development practice of “whole elements” of nature, society, and economy; the “whole space” of towns, agriculture, and ecology; and the “whole process” of past, present, and future developments.

The purpose of this study was to build a theoretical model of a green eco-city public service system from the three aspects of the construction criteria, influencing factors, and action mechanisms by applying the methodology of urbanology, ecology, and systems engineering to analyze its influence path and effect on green eco-city public service innovation by combining the environmental cognition of urban managers. The specific application of the research has two aspects. On the one hand, it can better construct an urban public service system under the background of current green and low-carbon developments and provide more green and healthy public services for urban residents. On the other hand, it can also help urban managers with green ecological environment cognition to discover the key and difficult problems in the provision of urban public services and to make targeted improvements.

## 2. Materials and Methods

### 2.1. Characteristics of Managers’ Environmental Cognition from the Perspective of Green Ecology

From the perspective of green ecology, city managers form different environmental cognitive characteristics based on different goal directions and management objectives, which will have different impacts on urban public service innovation. This study summarizes the composition characteristics, target characteristics, evolution characteristics, measures, and value characteristics of urban managers’ environmental cognition and explains the possible impact on urban public services (Table 1).

After analyzing and summarizing the environmental cognitive characteristics of city managers from the perspective of green ecology, this paper further analyzes and explains the structure of the house chart of environmental cognitive characteristics (Figure 1). As can be seen from the figure, complexity, systematicness, and openness constitute the basis for the formation of managers’ environmental cognition [5]. Dynamic and adaptive characteristics are necessary for managers’ environmental cognition [6]. Diversity, efficiency, complexity, safety, and health are the main goals for managers’ environmental cognition to be applied to green ecological city public services [7]. Circularity and compactness are the main measures for managers to provide green ecological city public services based on environmental cognition. Finally, symbiosis, human nature, and harmony are the ultimate value pursuits of urban management and urban public services provided by managers based on environmental cognition [8].

### 2.2. Connotations of Green Ecological City Public Services

Under the promotion of the concept of sustainable development, urban public services have also begun to advocate sustainable development. Meanwhile, sustainable urban public services pay more attention to improvement in the inclusiveness and equality of urban public services. However, in the sustainable development of urban public services, the climate, biology, energy, and economy are rarely involved. Therefore, on this basis, current city managers move out of the single urban public service category and consider a more global and multi-factor perspective, propose low-carbon city and low-carbon ecological city public services, plan the targets of urban public services more specifically, and add more factors for harmony between people and nature [9]. This has formed the basic prototype of a green city and ecological city public service in the process of urban management. Urban public service from a perspective of green ecology is the integration of a green city and an ecological city public service. No matter what the type of public service delivery method is, the ultimate goal is to achieve a perfect combination of an artificial environment, a natural environment, and a human environment in urban public services, ensuring the sustainable development of the city [10]. Based on the above discrimination, Figure 2 depicts the subordinate relationship of various urban public services from the perspective of sustainable development and green ecology.

As can be seen from the figure, both “green” and “ecological” in the public service of a green ecological city have been endowed with broader connotations. “Green” no longer reflects environmental factors, but is deeply combined with the connotation of sustainable development. Specific to the provision of urban public services, it means that urban public services are clean, safe, stable, and comfortable, with natural harmony and human health. “Ecological” is beyond the scope of the original ecology of a natural ecological system with traditional boundaries. From the phylogeny to the urban public services that provide an urban public service, there is an open working of the body that has complex and diverse characteristics. Urban public services provided by the target should have efficient coordination to achieve the city’s economic development and social wellbeing as well as an excellent ecological environment system.

### 2.3. The Theoretical Connection between Managers’ Environmental Cognition and Urban Public Services

After explaining the connotation of a green ecological city public service and the cognitive characteristics of managers’ environments from the perspective of green ecology, the theoretical basis of their influence and connection mainly comes from three theoretical pillars. One is the urban theory related to urban planning, including urban development theory, urban space theory, regional coordination theory, and urban economics theory. Second is the ecological sustainable development theory related to environmental ecology, including circular economy theory, compound ecology theory, ecological sustainable theory, and carrying capacity theory. The third is system theory, which mainly includes cybernetics, operations research, complex adaptive system theory, and system dynamics [11]. The specific theoretical basis is shown in Figure 3.

## 3. Analysis of the Results and Discussion

The public service innovation activity of a green ecological city is systems engineering. This chapter intends to simplify the expression of this complex system by constructing a theoretical model of a green ecological city public service system. Guided by the value objectives and development criteria of a green ecological city, and taking urban structure composition and organizational elements as objects supported by the logic of the urban development model and operation mechanism, it applies the principles, viewpoints, and methods of systems engineering to the research and solution of urban public service problems. It provides a conceptual model tool to analyze the impact of managers’ environmental cognition on urban public service provision; urban planning; management, construction and operation; and long-term sustainable development. The model constructed in this study also more clearly analyzed the specific aspects of the environmental cognition of city managers on green ecological city public service facilities under the background of green ecology, and through what methods.

### 3.1. Model Construction Criteria

A rule of model building is the goal of this research institute to build a model needed to achieve the results based on an analysis for managers handling the cognitive environment. The building green ecological perspective of the criterion is based on environmental cognition through city planning, construction, and operation management; the influence of these makes the city provide the public service that helps to promote a green ecological city construction [12]. Specifically, the criteria of this model include the general criteria and implementation criteria of a target set constructed at a general and sublevel. The construction criteria of this analysis model are shown in Figure 4.

It can be seen from the above figure that, from the perspective of green ecology, the general criteria of green ecological city public service model construction—namely, the macro-strategic goal—are sustainable development, coexistence, harmony, and being people-oriented. Managers must make their own management decisions by providing public services in the city, which achieve the economic constructions, political constructions, cultural constructions, social constructions and ecological civilization constructions of a “five one” all-round development through the mobilizing of all elements of nature, society, and economy in rural towns as well as the ecological whole space, taking into account the whole process of the past, present, and future and realizing the sustainable development of a harmonious coexistence between man and nature, society, and economy [13].

Furthermore, the model was mainly based on six specific implementation criteria to achieve specific classification objectives. Efficient circulation: In the implementation process of a green ecological city public service, there should be efficient circulation among the various subsystems. Various resources and energies can be fully utilized, and coordination between the various systems can be formed to achieve the most effective and reasonable utilization. Green health: Green ecological city public services should provide a safe and healthy urban environment, which can give full play to the creativity and productivity of people and maximize the protection of residents’ physical and mental health and living environment quality. The measures should be adjusted to local conditions; in a green ecological city construction of public services, obvious regional differences and characteristics of city managers will inevitably exist in the provision of public services and should fully consider the management of the city itself in terms of the geographical environment, cultural environment, economic environment, technology environment, and business and social environment as well as the differences and features [14]. The green ecological public service system constructed in different cities should be different rather than prevailing in different places and lacking differentiated public service systems. On the one hand, it is difficult to give full play to the fundamental goal of public service systems to serve a local green ecological city construction and the residents; on the other hand, it does not conform to the specific ecological conditions of local cities and other characteristics [15]. Compound diversity: The innovative development of green ecological city public services also needs to follow the principle of compound diversity. On the one hand, the diversification of urban public service forms to achieve the spatial integrity, continuity, and richness of urban public services. On the other hand, the concept of green ecology should be fully upheld to ensure the diversity of humans and nature in urban life and accommodate diverse people and natural creatures [16]. Coordination and mutual promotion: Green ecological city public services should also contribute to coordinated and mutual development among urban residents, the surrounding natural environment, and the local economy as well as society and the livelihood of people. Especially under the environmental cognition of green compound ecology, urban management needs to pay more attention to the benign interaction and development of economic systems, social systems, and natural systems. Inclusive and open: In the complex system of a green ecological city, which is filled with various elements such as the capital, resources, and information, public services from the perspective of green ecology cannot be analyzed from a closed and narrow perspective. The analysis should be carried out with an open attitude, and a diversified and composite open system should be created, which coexists with the natural environment where organisms exist equally and mutually, activities are convenient, and the values of greenness and inclusivity should be advocated.

### 3.2. Main Influencing Factors

After analyzing the construction criteria of the green ecological city public service system theoretical model combined with the classification of urban managers’ environmental cognitive characteristics, we further analyzed the influencing factors on green ecological city public services. The relevant influencing factors were selected and analyzed from the perspective of urban management, centering on the development requirements of urban social, economic, and environmental systems and under the construction principle of a green ecological city public service system model.

For the study of the influencing factors, there were three main factors for different research purposes. One was to divide cities into natural, economic, and social aspects according to the management objectives and discuss the role of managers’ environmental cognition on their public services from three aspects [17]. The first category was classified from the characteristics of the public services themselves; the differential influence of managers’ environmental cognition was studied according to different types of public services [18]. The other one took the city as an ecosystem and analyzed the role of managers’ cognition from the aspects of the system structure, function, and coordination [19].

This study draws on the analysis experience of previous research viewpoints and discusses the impact of managers’ environmental cognition on urban public services from five aspects, including social status, economic status, natural environment, resources, and population whilst taking the city as a whole ecosystem. In other words, from the perspective of green ecology, managers’ environmental cognition mainly affects urban public services through the above five aspects.

According to the above analysis and the theoretical basis of the connotation of the public service system of a green ecological city, we considered which specific variables would make further responses or changes to the above factors. In a public service system, basic public service industry, ecological protection public service, or public space or site, the public cultural facilities, transportation, social security public service, green science, technology public service, and urbanization construction public service will be affected. According to the function hierarchy and order of the above five factors, the main influencing factors of managers’ environmental cognition on urban public services from the perspective of green ecology were constituted [20]. The composition relationship of the specific influencing factors is shown in Figure 5.

### 3.3. Mechanism of the Model

After obtaining the main influencing factors, to build a green ecological urban public service system theory model managers need, through the analysis of the cognitive environment, the specific role of the green ecological city public service innovation mechanism through to the internal organization structure and interaction between the influence factors and the method of analysis. This enables the connected factors to form a final analysis model. Figure 6 depicts the main path of action.

As can be seen from the figure, the mechanism of the impact of managers’ environmental cognition on green eco-city public services is a model where the logic of the whole process of the action mechanism is formed by taking the macro-background and theoretical support as the external framework and the operation, construction, planning, renewal, capital, and governance as the internal components. The driving forces, management mechanisms, and financial support are the auxiliary components.

External framework: In terms of the macro-background, the process of urban management includes the economic conditions, cultural structure, national characteristics, geographical environment, historical tradition, social form, and other time characteristics of the urban residents in a specific macro-background. The research on the impact of green ecological city public services also needs to be carried out under a specific background and macro-environment. Theoretical support: The theory technology level also affects and restricts the city development; for the urban planning design and construction of public services management, with enough theory as the guidance, the city builders and managers’ cognitive connotation of city development can decide their ability to solve the problem of urban development for different countries and regions with different natures and scales as well as the different stages and cultural cities. The theoretical basis supporting and guiding urban development is also completely different, which requires reasonable screening and selection [21].

Internal composition and driving force: the sustainable development of a city is based on the results of the joint action of various driving forces, which are mutually conditioned and coordinated. Various dynamic factors not only reflect the quality of urban development, but also enrich and transform the development of urban politics, economy, society, and humanity. The rapid development and innovation of urban public services is also inseparable from the creation and renewal of internal compositions and driving forces, so governments need to manage the city from an operational, construction, planning, renewal, capital, and governance point of view along with other aspects. At the same time, from the perspective of green ecology and based on the concept of development, the urban management of city managers should be carried out based on the value goals of environmental cognition. The main body of these urban management activities is not completely the government, nor is it completely dependent on administrative means and other coercive forces. Urban management is the cooperation between governments and urban people, government and non-governmental organizations, public and private institutions, and compulsory and voluntary cooperation. The provision of urban public services is also a process of top-down interactions. The government, non-governmental organizations, and various private institutions deal with public affairs through common goals based on diversified means such as cooperation and consultation, so their power dimensions are diversified rather than pure top-down. Therefore, the development and innovation of public services in a green ecological city is a process in which the government, together with society and other forces, participate in and act together as the driving force for development [22].

Auxiliary means: Managers’ environmental cognition also needs the existence of auxiliary mechanisms to realize the mechanism of urban public services, which involves two aspects. On the one hand is the management system. The provision of urban public services and policy operation cannot be separated from the institutional guarantee, especially the urban administrative system, which is based on the economic system and adapts to the current economic system, determining the functional development and positioning of the city. With an economic system, it is important to note the other side of auxiliary means; namely, urban financial support. Urban financial support is the guarantee for a successful implementation of the various measures and means. The system and scale of urban financial revenue and expenditure directly restrict and influence the speed and quality of urban public service development. The construction of a green ecological city, the guarantee of a balanced, equal, reliable, and adaptive provision of public services, the adjustment of urban income distribution and urban economic development, and the improvement of the urban governance capacity are inseparable from the support of urban finance.

The logic mode of the mechanism of action: First of all, managers’ environmental cognition affects urban public services in a whole process and managers’ influence on green ecological city public services is a whole process practice. Meanwhile, in the practice process, managers should fully combine their own local characteristics and various adjustment differences to obtain a reasonable layout. Therefore, its function is the fusion of a variety of factors through a long period of effect evolution and formation. It is also a balanced path and self-sustaining system to observe the whole process of urban management and public service.

Time effectiveness: Managers also differ in their cognition of green ecology at a time level because the provision of public services is a whole process practice that includes planning, construction, and governance. According to the different stage requirements, managers should give play in an orderly, successive, and phased manner. They should recognize the effects of policy guidance, resource support, and cultural dominance in the early stage; capital drive, market construction, and industrial development in the middle stage; and green ecological location drive and value concept dissemination in the late stage to give full play to the improvement of intergenerational distribution efficiency based on policy recognition.

In general, through the analysis of each part of the model’s mechanism, the paper analyzed the operation logic and rules of manager cognition in urban public service innovation, which has a certain stage and reference significance. At present, the values of urban management in the world are changing from the direct pursuit of economic efficiency maximization to the pursuit of social fairness and justice; from materialism to the protection and production of cultural and ecological values. It can be said that the research on the innovation and development of green ecological city public services in this paper came into being with the progress of urban management and environmental changes, which could provide a reference for the development and management of public services in other cities.

### 3.4. Model

Based on the analysis and construction of the construction criteria, main influencing factors, and mechanism mentioned above, this study comprehensively established a system model describing the impact of managers’ environmental cognition on urban public service innovation and development, as shown in Figure 7. Based on this model, feasible concrete measures and influencing factors can be analyzed through the orientation of city managers for green ecological cities in terms of objectives, functions, and operation mechanisms. This paper provides a three-dimensional and comprehensive overview of the green eco-city public service, a complex whole system, and provides a good method and tool for green eco-city management.

## 4. Conclusions

Through the research, it was found that the theoretical basis of green ecological city public services mainly came from the three pillars of urbanology, ecology, and systematics. By using the above theories and a practical analysis, the following understandings were obtained. Managers’ environmental cognition from the perspective of green ecology is characterized by systematization, openness, and complexity. The main goal of management is diversification, efficiency, compounds, safety, and health. In the management process, there are the development laws of being dynamic and adaptive. The management measures taken are mainly circular and compact, and the value goals of symbiosis, being people-oriented, and harmony are realized by acting on urban public services. A green ecological city public service is a comprehensive and complex system. The extension of “green” and “ecological” concepts requires more scientific, more systematic, and more natural management methods to provide public services. The environmental cognition that urban managers should have include being systematic, open, and complex. Urban management should pursue diversification, efficiency, safety, and health. The theoretical model of a green ecological city public service system constructed in this study analyzed and explained the relationship between managers’ behavior and urban public service innovation and development in a relatively three-dimensional and comprehensive way, providing a reference for a better understanding of the system composition as well as the innovation and development laws of green ecological city public services.

## Figures and Tables

**Figure 1 ijerph-19-15945-f001:**
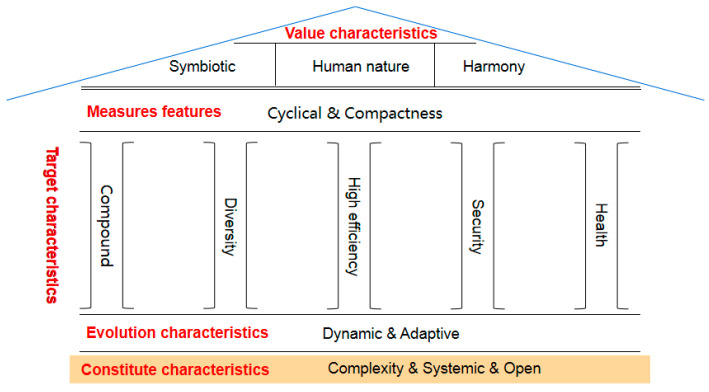
House diagram of environmental cognition characteristics of urban managers.

**Figure 2 ijerph-19-15945-f002:**
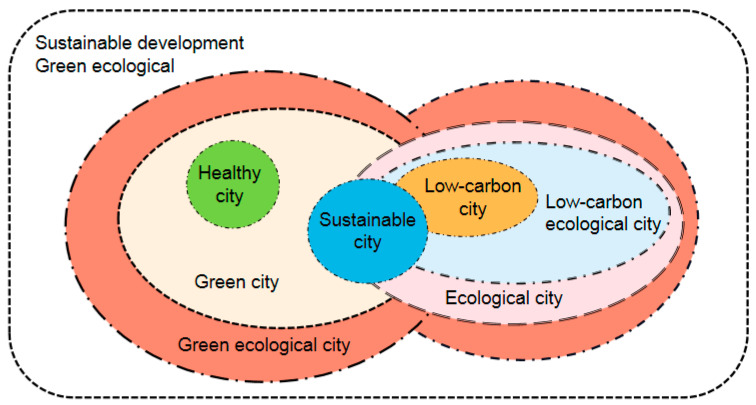
The subordination relationship of various urban public services from the perspective of green ecology.

**Figure 3 ijerph-19-15945-f003:**
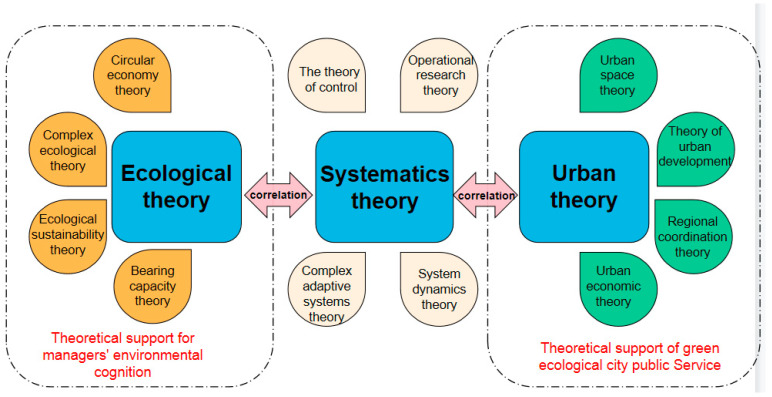
Composition of theoretical connection.

**Figure 4 ijerph-19-15945-f004:**
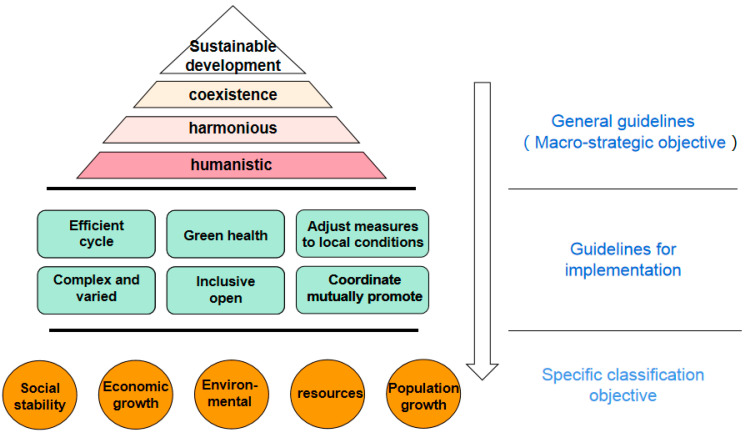
Partition of model construction criteria.

**Figure 5 ijerph-19-15945-f005:**
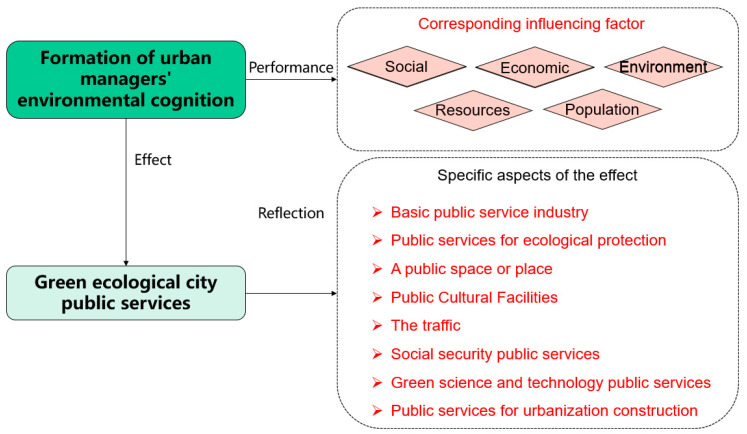
Composition of major influencing factors of managers’ environmental cognition on public service innovation.

**Figure 6 ijerph-19-15945-f006:**
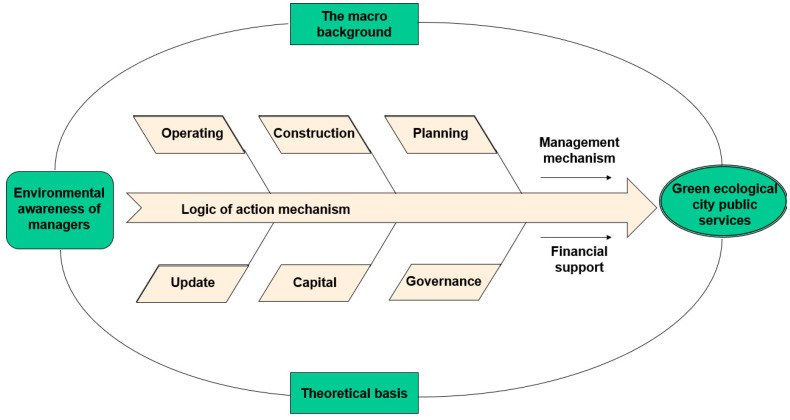
Managers’ environment cognition and the mechanism of action of the public service of a green ecological city.

**Figure 7 ijerph-19-15945-f007:**
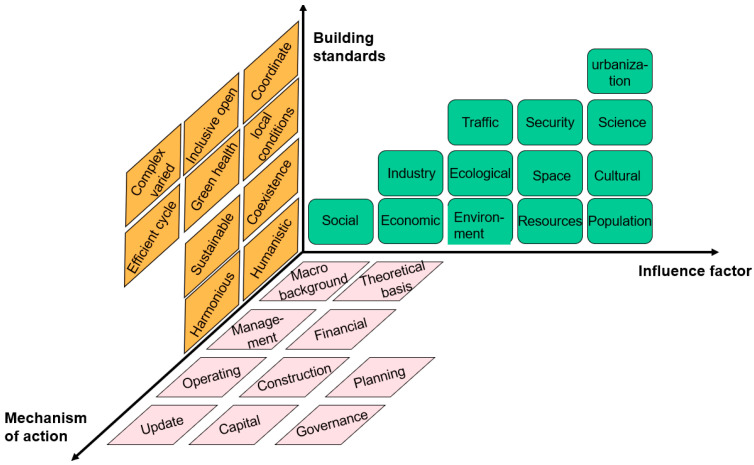
Green ecological urban public service system theory model.

**Table 1 ijerph-19-15945-t001:** Environmental cognitive characteristics of urban managers.

Characteristics of Categories	Specific Characteristics	Explanation
Constituent Characteristics	System	The composition of managers’ environmental cognition is systematic. As city management involves many elements, their environmental cognition must seek the balance of the whole system in multi-dimensions.
	Open	The process of urban management is a process of exchange and replacement with the outside world, so its environmental cognition must ensure that the city always maintains an open circulation and circulation with the outside world.
	Complexity	Environmental cognition is complicated because the core purpose of both urban management and public services is to serve urban residents. The complexity of human beings results in the complexity of environmental cognition.
Target Characteristics	Diversity	The management of a green ecological city should show diversity in levels, elements, and associations to better guarantee the diversity and vitality of the city.
	High Efficiency	From the perspective of green ecology, urban management should be efficient to achieve sustainable urban development.
	Compound	In urban management, the green concept and ecological concept are fully combined to realize the coordination and unity of ecological, social, production, cultural, and other urban functions.
	Security	Safety is an important prerequisite for management. From the perspective of green ecology, managers’ environmental cognition goals should also pay attention to the security guarantee of urban social stability, economic development, natural resources, the environment, public safety, and health.
	Health	Urban management should contribute to the health of urban residents, their living environment, and their future development.
Evolution Characteristics	Dynamic	As things develop, the environmental cognition formed by managers should also have a certain dynamic and it should dynamically adjust with the process of urban development.
	Adaptive	The urban system responds to the external environment and eventually feeds back into the human and social systems. Therefore, managers’ environmental cognition also needs to adapt to this feedback and achieve the dynamic adaptation of the internal and external circulation of management through continuous adjustment, feedback, re-adjustment, and re-feedback.
Features of the Measures	Cyclical	From the perspective of green ecology, the measures taken by managers based on environmental cognition should reflect the concept of circular development, which is a virtuous cycle of all systems and levels of the city, to improve the overall urban public service efficiency and resource utilization rate.
	Compactness	Compactness refers to the specific measures formulated by managers to pay attention to the appropriate and compact use of injected land, public infrastructure, and public services rather than ignoring costs in order to achieve green ecological goals.
Value Characteristics	Human Nature	The value of urban managers’ environmental cognition is primarily people-oriented. Whether it is the construction of green ecological cities or the provision and innovation of public services, the fundamental goal and the method of value realization are to realize the comprehensive and free development of human beings on the basis of a harmonious coexistence between man and nature.
	Symbiotic	Symbiosis emphasizes that managers should pay attention to the city rather than a single individual in their environmental cognition. In the process of providing green and ecological public services, they should pay attention to mutual assistance and mutualism outside the city, between cities and regions, and between cities.
	Harmony	It is emphasized that managers’ environmental cognition should be based on harmony. It is not necessary to sacrifice the interests of a small number of groups to achieve a certain goal, but all urban residents should do their best and give full play to their strengths whilst enjoying the wellbeing brought about by green ecological cities so that harmony and consistency can help urban development.

## Data Availability

The labeled dataset used to support the findings of this study is available from the corresponding author upon request.

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
