# Peer review of "The Impact of Managers’ Environmental Cognition on Urban Public Service Innovation from the Perspective of Green Ecology"

_ijerph, 2022, doi:10.3390/ijerph192315945_

Round 1

Reviewer 1 Report

1. This paper proposes that city managers should use the environmental cognition of green ecology as the basis when dealing with public services. The important conclusions put forward: the environmental cognition that urban managers should have been systematic, open and complex. The goals of urban management are diversification, efficiency, complexity, safety, and health. The methods of urban management include circularity, compactness, harmony, humanistic orientation, and symbiosis.

2. The cognitions and concepts proposed in this paper should belong to the policy level, and it is suggested that they can be integrated into the indicator system at the practical level.

3. Urban management in addition to policy cognition at the spatial level, policy cognition at the time level for resource allocation is also an important criterion. It is recommended to discuss the inter-period and inter-generational allocation efficiency, equity, and stability strategies of supplementary resources.

Author Response

1.This paper proposes that city managers should use the environmental cognition of green ecology as the basis when dealing with public services. The important conclusions put forward: the environmental cognition that urban managers should have been systematic, open and complex. The goals of urban management are diversification, efficiency, complexity, safety, and health. The methods of urban management include circularity, compactness, harmony, humanistic orientation, and symbiosis.

Answer: Thanks. This is just what this paper intends to expound and indicate.

2.The cognitions and concepts proposed in this paper should belong to the policy level, and it is suggested that they can be integrated into the indicator system at the practical level.

Answer: In the part 3.3 of model mechanism, according to the modification requirements and combined with Figure 6, it explains the logical thinking of how the cognitive concept influences the practical application level based on the policy level. At the same time, in chapter 3.4 of the paper, also added the introduction of this aspect from the policy to the actual explanation.The corresponding additions are marked in red.

3.Urban management in addition to policy cognition at the spatial level, policy cognition at the time level for resource allocation is also an important criterion. It is recommended to discuss the inter-period and inter-generational allocation efficiency, equity, and stability strategies of supplementary resources.

Answer: According to the suggestions, in Chapter 3.3, the corresponding influence of policy cognition on the time level and the specific application mode are added, and the time level of policy cognition is divided into before, during and after the provision of urban public services, which provides a reference for improving the efficiency of intergenerational distribution during the time period.

Reviewer 2 Report

The article presents an interesting theoretical model on the impact of managers' environmental cognition on urban public service innovation from the perspective of green ecology. The article has a definite and logical workflow, supplemented with clear illustrations presenting the discussed results. The methodology is well illustrated. Discussion in the paper of the local government unit is well-grasped and the conclusions result from the manscript. However, the article itself is very conceptual and even too much and requires a clear research objective and the ability to apply the concept to concrete examples. For me as a practice in the field, it is ultimately not clear why the article could be advantageous in practice. The use of the concept  justifies the inclusion of the topic and a slightly more comprehensive literature review.

What contribution do the authors of the paper make to urban ecology?

The work contains editorial errors, e.g. the labeling under some Figures starts in lowercase letters. Ultimately, I think the work needs a minor revision before it is submitted for publication in IJERPH MDPI Journal.

Author Response

1.The article presents an interesting theoretical model on the impact of managers' environmental cognition on urban public service innovation from the perspective of green ecology. The article has a definite and logical workflow, supplemented with clear illustrations presenting the discussed results. The methodology is well illustrated. Discussion in the paper of the local government unit is well-grasped and the conclusions result from the manscript. However, the article itself is very conceptual and even too much and requires a clear research objective and the ability to apply the concept to concrete examples. For me as a practice in the field, it is ultimately not clear why the article could be advantageous in practice. The use of the concept justifies the inclusion of the topic and a slightly more comprehensive literature review.

Answer: According to the requirements of the review, the research purpose of this paper is supplemented, which is to provide a "theoretical model of green ecological city public service system" and analyze the effect of urban managers' environmental cognition based on the model. The specific application of the research is mainly to provide theoretical guidance for the construction of green urban public service system and urban managers to solve the key difficulties in the process of providing public services.In the first chapter of the article, the corresponding explanatory content is added, marked with red font.

2.What contribution do the authors of the paper make to urban ecology?

Answer: The main contribution of this study is to provide an urban public service construction and management system based on urban managers' own environmental cognition, which provides theoretical guidance for urban public service supply under the current green ecological background. The specific contributions are supplemented at the end of Chapter 1.The additions are indicated in red.

3.The work contains editorial errors, e.g. the labeling under some Figures starts in lowercase letters. Ultimately, I think the work needs a minor revision before it is submitted for publication in IJERPH MDPI Journal.

Answer: Thanks for the reviewer's suggestion, the digital title content details of the full text have been checked and corrected.